



# Doppio – A ROMS-based Circulation Model for the Mid-Atlantic Bight and Gulf of Maine: Configuration and comparison to integrated coastal observing network observations

Alexander G. López[1], John L. Wilkin[1], Julia C. Levin[1]

[1]Department of Marine & Coastal Sciences, Rutgers, The State University of New Jersey, New Brunswick, NJ 08901, United States of America

*Correspondence to*: Alexander G. López (alopez@marine.rutgers.edu)

**Abstract.** We describe "Doppio", a ROMS-based model of the Mid-Atlantic Bight and Gulf of Maine regions of the northwest North Atlantic developed in anticipation of future applications to biogeochemical cycling, ecosystems, estuarine downscaling,

and near-real-time forecasting. This free-running regional model is introduced with circulation simulations covering 2007-2017. The ROMS configuration choices for the model are detailed, and the forcing and boundary data choices described and explained. A comprehensive observational data set is compiled for skill assessment from satellites and in situ observations from Regional Associations of the U.S. Integrated Ocean Observing Systems, including moorings, autonomous gliders, profiling floats, surface current measuring coastal radar, and fishing fleet sensors. Doppio's performance is evaluated with respect to these observations

by representation of sub-regional temperature and salinity error statistics, as well as velocity and sea level coherence spectra. Model circulation for the Mid-Atlantic Bight and Gulf of Maine is visualized alongside the mean dynamic topography to convey the model's capabilities.

## 1 Introduction

Coastal ocean circulation models that downscale global ocean simulations are useful tools for exploring regional ocean dynamics

and associated links to biogeochemistry, ecosystems, geomorphology, and other applications; for example, by inferring transport pathways for nutrients, larvae, sediments, or pollutants. The reduced geographic scope of a regional model offers economies in computational effort that allow much greater experimentation than would be possible with global models alone, as by examining sensitivity to resolution or parameterization of added physics, and they present the opportunity to affordably explore numerous application scenarios. Here we describe the development, evaluation, and application of a regional model of the northeast

continental shelf of North America from Cape Hatteras, North Carolina, northward to near Halifax on the Scotian Shelf of Canada. The model, intended principally for studies of ocean physical circulation but conceived for future applications to biogeochemical cycles and ecosystems, uses the three-dimensional hydrostatic shelf circulation model ROMS (Regional Ocean Modeling System; www.myroms.org) [*Shchepetkin and McWilliams*, 2005] as the underlying hydrodynamic model core.

The model configuration builds significantly on two earlier regional modeling programs. A ROMS Northeast North American (NENA) shelf coupled circulation and biogeochemical model encompassing the entire coastal ocean extent from Florida to the



Grand Banks of Newfoundland [*Hofmann et al.*, 2008] was used for numerous studies of nutrient and carbon fluxes in this region [*Fennel and Wilkin*, 2009; *Fennel et al.*, 2006; *Fennel et al.*, 2008]. The NENA biogeochemical model performed well within the Mid-Atlantic Bight but less so for the Gulf of Maine [*Hofmann et al.*, 2008], and this lackluster performance in the

Gulf of Maine was ascribed to shortcomings of the modeled physical circulation [*Cahill et al.*, 2016]. Accordingly, an emphasis in configuring the model described here was to create an improved Gulf of Maine circulation so that subsequent biogeochemical simulations will have a more skillful physical foundation. A second prior ROMS-based modeling effort, termed ESPreSSO (Experimental System for Predicting Shelf and Slope Optics), had a more limited geographic scope covering only the Mid-Atlantic Bight [*Zavala-Garay et al.*, 2014]. This model has been widely used for studies ranging from hurricane-induced cooling

via mixing [*Seroka et al.*, 2017] to shelf-wide ecosystems [*Xu et al.*, 2013] and dissolved organic carbon fluxes [*Mannino et al.*, 2016]. An operational forecast version of ESPreSSO that used 4-dimensional variational assimilation [*Levin et al.*, 2018; *Zavala-Garay et al.*, 2014] performed the best of seven real-time models covering the region [*Wilkin and Hunter*, 2013].

The present modeling effort, which we have dubbed "Doppio", focuses on maintaining the skill shown by ESPreSSO while expanding the domain to include the Gulf of Maine. To assess the "Doppio" skill the observing network used for ESPreSSO was

expanded, adding new satellite altimeters and SST sensors and the Gulf of Maine in situ observations.

The moniker "Doppio" reflects that the Doppio domain is approximately twice the extent of ESPreSSO. The model domain is indicated in Fig. 1, colored by bathymetry and with positions marked for several time series observation locations used for either forcing or analyses that will be discussed later in the paper.

50                                        [Figure 1]

The Doppio domain encompasses two very different dynamical regimes in the Mid-Atlantic Bight (MAB) and the Gulf of Maine. The MAB (Cape Hatteras, North Carolina to Cape Cod, Massachusetts; Fig. 1) is characterized by a broad (~100 km-wide) shelf with a permanent front at the shelf-break that separates relatively cool and fresh shelf waters from the warmer and

more salty Slope Sea [*Mountain*, 2003]. Instabilities in the shelf-break front have wavelengths of typically 40 km that can evolve significantly in time over just a few days [*Fratantoni and Pickart*, 2003; *Gawarkiewicz et al.*, 2004; *Linder and Gawarkiewicz*, 1998]. The along-shelf currents generally reach the seafloor, exhibiting significant flow-bathymetry interactions and establishing across-shelf transport in the bottom Ekman layer.

Eddy shelf interactions tied to Gulf Stream-induced warm core rings [*Zhang and Gawarkiewicz*, 2015] lead to cross-shelf exchange with surface and sub-surface structure at scales of 10-30 km and days to weeks. Subsequent across-shelf fluxes of heat, freshwater, nutrients, and carbon control water mass characteristics and impact ecosystem processes throughout the MAB.

The Gulf of Maine (GOM) is a semi-enclosed marginal sea distinctive in the world for having the largest tidal amplitude, over 16

m, due to its shape and length that lead to near resonance of the lunar semi-diurnal $M_2$ constituent of the tide [*Garrett*, 1972]. There are two main oceanic inflows to the Gulf of Maine: Scotian Shelf water flowing southwestward along the coastline from Halifax and originating from the Labrador Current; and Slope Sea water entering through the Northeast Channel that derives from subpolar North Atlantic waters mixed with eddies of the Gulf Stream; additional inflows are river runoff from many sources along the coasts of New England, New Brunswick, and Nova Scotia; and the net difference in precipitation and

evaporation within the Gulf of Maine [*Brown and Beardsley*, 1978]. The two main outflows are water exiting through the Great



South Channel between Cape Cod and Georges Bank toward Nantucket, and around Georges Bank [*Brown and Beardsley*, 1978]. This exchange flow through the Northeast Channel can be influenced by Gulf Stream eddies, episodically delivering warm, saline waters [*Bisagni and Smith*, 1998] in such quantity as to change the physical circulation of the Gulf of Maine [*Brooks*, 1987]. The circulation is predominantly counter-clockwise about the Gulf of Maine, from Nova Scotia into or across the

Bay of Fundy, then in a strong coastal current southward along the New England coast. While some water exits via the Great South Channel, the majority of flow proceeds eastward along the northern flank of Georges Bank, finally wrapping around the underwater plateau and continuing towards the MAB [*Bigelow*, 1927; *Wiebe et al.*, 2002]. Within the Gulf, strongly irregular bathymetry exerts significant control on the low frequency flow variability above three deep basins, which can be challenging to model as previous studies have shown [*Hofmann et al.*, 2008]. The Gulf's confluence of uncommon bathymetry and strong tidal

forcing lends itself to equally uncommon currents, namely a significant along-bank current jet that may be the prime driver of transport through the region [*Loder et al.*, 1992].

Physical circulation processes influence the biogeochemistry of the Gulf of Maine via a number of mechanisms. Hibernal circulation is especially dynamic, influenced by winds on short time scales, and partial mixing of three separate water masses

[*Vermersch et al.*, 1979]. Mixed layer depth influences the onset of primary productivity via vernal mixing, with shallower regions of approximately 60 meters or less conducive to more substantial and sustained productivity [*Yentsch and Garfield*, 1981]. Recent warming has resulted in increased rainwater entering the gulf, freshening the surface and stratifying the water column, inhibiting vertical nutrient flow [*Salisbury et al.*, 2009]. Within the Gulf, strong estival recirculation causes retention of both primary producers and nutrients for no less than forty days [*Naimie et al.*, 2001]. Improving our capability to model the

physical circulation of this region, and to determine what may be controlling carbon air-sea exchange and reservoirs at a regional level is important to developing a full comprehension of the carbon cycle at the global scale.

## 2 Model configuration for the MAB and GOM

Our regional model is created using ROMS, a three-dimensional hydrodynamic model that solves the hydrostatic, Boussinesq, primitive equations in a structured horizontal grid with terrain-following vertical coordinates. The ROMS computational design

itself and many of the model's companion features such as integrated sediment transport and ecosystem/biogeochemical models are described in detail elsewhere [*Haidvogel et al.*, 2008; *Shchepetkin and McWilliams*, 2005; 2009]. ROMS is used extensively for coastal and continental shelf applications.

The Doppio model, building on the ESPreSSO heritage, uses many of the same model settings and parameter values. The model

resolution is a uniform 7 km horizontal grid (242 x 106 cells) and 40 vertical levels. This resolution is a compromise, as a finer horizontal resolution would help capture submesoscale dynamics but would dramatically increase the computational costs. Given the multitude of model runs to be undertaken during model configuration and then application, it is practical to employ the modest 7 km uniform resolution. In comparison, ESPreSSO had a 5 km grid resolution and 36 vertical levels, and NENA had a 10 km grid resolution and 30 vertical levels but also covered the entire Gulf of Maine, MAB, and South Atlantic Bight. The

vertical stretching is such that the resolution is enhanced toward surface and bottom boundary layers in the coastal ocean (inside the 100 m isobath), and there is better than 3 m resolution at the seafloor and 1.5 m resolution at the sea surface. Vertical mixing is parameterized using the generic length-scale (GLS) [*Umlauf and Burchard*, 2003] implementation of the k-kl turbulence





closure [*Mellor and Yamada*, 1982]. A detailed listing of other configuration options and parameter choices is presented in Table 1.


The Doppio model configuration has been applied to simulations of the decade 2007-2017. Over this period we have reliable and consistent meteorological forcing and open boundary condition data, and a dense set of observations with which to assess the model skill. The locations of river point sources used for forcing, along with tide gauges and moorings used for the skill assessment, are noted in Fig. 1. The model has also been implemented, essentially unchanged, as an experimental operational

ocean forecasting system with variational data assimilation. The forecast system is not a focus of the present study, but several of the choices of model input data streams were motivated by the intent to allow near-real-time operation. To complete the description of the model configuration we detail next the external driving data sets that determine the air-sea fluxes, river inflow, and open boundary forcing.

120                                     [Table 1]

### 2.1 Atmospheric Forcing

Atmospheric forcing data are drawn from National Centers for Environmental Prediction (NCEP) products, namely the North American Regional Reanalysis (NARR) [*Mesinger et al.*, 2006] and North American Mesoscale (NAM) [*Janjic et al.*, 2005] forecast model. The atmospheric analysis variables used are net shortwave and downward longwave radiation, precipitation, and

marine boundary layer air pressure, temperature, relative humidity, and wind velocity. With these and model sea surface temperature the air-sea fluxes for momentum and heat are calculated according to the so-called TOGA-COARE bulk fluxes parameterization [*Fairall et al.*, 2003]. The air pressure also directly drives sea level variability via the Inverted Barometer effect (ATM_PRESS in Table 1).

An essential atmospheric forcing term is net shortwave radiation flux (downwelling shortwave radiation minus the fraction reflected due to ocean surface albedo), which is important not only for its influence on model physics but also as a driver of primary productivity when circulation is coupled to models of ocean biogeochemistry and ecosystems. It has been noted in previous studies that NARR shortwave radiation tends to be an overestimation in comparison to observed values [*Kennedy et al.*, 2011], and a study within our region of interest, namely Delaware Bay [*Wang et al.*, 2012], applied a reduction of NARR

shortwave by 20 % to correct for this (though the analysis actually showed the overestimation to be typically 23 %). To examine whether a 23 % correction is warranted beyond Delaware Bay, we compared net NARR shortwave to weather satellite radiance observations from ISCCP (International Satellite Cloud Climatology Project) [*Schiffer and Rossow*, 1983] at one point, the ground station MVCO (Martha's Vineyard Coastal Observatory), and observe in Fig. 2(a) an overestimation by NARR of 17 %. The ISCCP spatial resolution is low compared to local land-based radiometer data in *Wang et al.* [2012] so, to further test the

*Wang et al.* [2012] analysis, we compared to a higher-resolution satellite product in the form of downwelling Photosynthetically Available Radiation (PAR) from MODIS (http://coastwatch.pfeg.noaa.gov/erddap/griddap/erdMEpar01day.html) [*Van Laake and Sanchez-Azofeifa*, 2005]. In Fig. 2(b) we have applied the 23 % adjustment to NARR net shortwave, converted net to downward shortwave assuming an albedo of 6 % [*Payne*, 1972], and applied a factor of 0.45 [*Kirk*, 2010] for the fraction of



shortwave that is PAR for comparison to MODIS. We see that the mean ratio of the two is approximately 1 and are reassured

that the NARR reduction of 23 % is justified.

[Figure 2]

The shortwave radiation values from NARR and NAM are instantaneous diagnostic quantities calculated from the modeled water

vapor and other atmospheric constituents, and are provided at three-hour intervals. This interval poorly resolves the diurnal cycle

of air-sea heat flux and, potentially, photosynthesis.

The time of solar noon varies across the longitude extent of the domain and shifts with respect to the reporting hour during the

seasonal cycle. It can be shown that the clear sky maximum radiation reported with a 3-hour sampling interval is typically

underestimated by 5% but can be underestimated by as much as 20% when solar noon falls between the 3-hour samples.

Therefore, to better capture the full range of the solar cycle, daily averages of the NARR or NAM data are provided to the model

and at runtime an idealized diurnal shortwave radiation cycle is imposed, appropriate to the longitude, latitude, and year day, that

has the same daily average (DIURNAL_SRFLUX in Table 1). This option ensures the correct length of day and better noontime

peak solar radiation.

### 2.2 River Sources

The Gulf of Maine and Mid-Atlantic Bight are home to many rivers with moderately high discharge that varies on quite short

time scales, from the Saint John River entering the Bay of Fundy in Canada, all the way down to the Susquehanna River entering

the Chesapeake Bay, Maryland. A significant fraction of this terrestrial runoff makes its way to the ocean without joining a river

that is actively monitored by the United States Geological Survey (USGS) or the Water Survey of Canada (WSC). To account

for flow that reaches the GOM and MAB via ungauged portions of the watershed, we turned to a high spatial resolution

watershed model based analysis of daily river discharge that aggregated surface water flow into a total of 403 rivers along the

northeastern seaboard of North America for the eleven-year period (2000-2010) [*Stewart et al.*, 2013]. This represents a near

complete accounting of the terrestrial surface water discharge from land to ocean. These 403 modeled sources were consolidated

within large watersheds into 27 principal river sources; 24 in the United States and 3 in Canada (Fig. 3). The locations along the

coast of the discharge-weighted consolidations were, in most cases, associated to one large familiar named river source. The

consolidated data set therefore comprises a decade-long record of daily total watershed discharge for a modest number of river

sources suited to driving the regional ocean model. However, the retrospective analysis time period 2000-2010 leaves us without

river forcing data for subsequent years. The locations of point sources contributing to each of the 27 consolidations were again

compared with watershed maps to find the USGS and WSC river gauges nearest to the mouth of the chosen rivers that are known

to reliably report daily data in near real time.


[Figure 3]

A second data set comprising these 27 river gauges for the 2000-2010 interval was compiled and a maximum covariance analysis

(MCA) of the two data sets was used to formulate a predictor of full watershed discharge at the 27 principal river sources based

on the USGS and WSC gauge data as inputs. This statistical extrapolation compensates for the ungauged watershed and we use





this in place of the time span limited *Stewart et al.* [2013] data set, or the gauge data alone, both of which are incomplete for our purposes. Furthermore, the method is suited to use in near real time by operationally acquiring the gauge data and applying the statistical expansion to the full watershed discharge.

An obstacle to this approach, however, is gaps in the WSC data for both the real-time and historical sections. Additionally, there are no real-time river gauges in Nova Scotia, Canada, that report discharge; the only parameter reported is water level. Therefore, a rating curve approach was taken by computing a relation between water level and discharge for the Mersey River, one of the two Nova Scotian stations in question, using historical data. Discharge data for Mersey River is available through 2012, while the station began recording water level in early 2011 and continues to do so operationally. Figure 4(a) shows water level and

discharge for the period 2011–2012 when simultaneous data exist, and the quadratic relation inferred to project discharge on the basis of water level data.

[Figure 4]

In Fig. 4(b) the projection (red) is made for the entirety of the water level record, showing strong correspondence between

measured and inferred discharge in the overlap period giving credence to the relation as a useful predictor of discharge from the real-time water level. There was no comparable training data set for the Sackville River in Nova Scotia, but an analysis of correlations between Sackville River discharge and all neighboring rivers showed the Mersey (128 km away) as a likely predictor. Though the correlation in discharge is not particularly strong (Fig. 4(c)), using the Mersey river to project discharge for the Sackville during times of coincident data availability (Fig. 4(d)) captures the timing and magnitude of peak discharge

events well, giving a useful real-time discharge predictor based on river water level data. A 10-month gap in USGS data for the Carmans River, New York, was filled following a comparable process utilizing a discharge and water level relation for the nearby Quashnet River. Other minor data gaps of order a few days or so were simply filled by linear interpolation.

Water temperature is reported at only few of the discharge stations, and is often incomplete at best. Therefore, in lieu of direct

observations, the river temperatures provided to the ROMS model were interpolated from NARR atmospheric forcing data but capped at 0° C minimum. The mean temperatures are indicated in Fig. 3. Where gauge data were available for comparison, the NARR temperatures were on average a few degrees warmer than the gauge temperatures. This is inconsequential in the ocean response because at discharge locations the water temperature is quickly modified by mixing and air-sea heat fluxes.

With these processing steps complete, we have discharge data for 27 principal rivers along the eastern seaboard, stretching from Nova Scotia down to the Carolinas for 2007 through 2019 to drive the hind-cast regional ocean circulation experiments described below. Moreover, the methodology has been adapted to run the system in near real-time for operational ocean forecasting.

### 2.3 Open Boundaries

Open boundary information at the model domain perimeter is based on daily mean data taken from the Mercator-Océan system [*Drévillon et al.*, 2008] provided by Copernicus Marine Environment Monitoring Service (CMEMS; marine.copernicus.eu). To these data we apply a bias correction to the annual mean, retaining mesoscale variability. These corrections are substitutions of





temperature and salinity with the annual mean from our regional climatological analysis MOCHA [*Fleming*, 2016] and Mean

Dynamic Topography and velocity are from data-assimilative climatological analysis with the Doppio grid. While others have

shown that sourcing open boundary data from global products, even with a bias correction, may not always yield better results

than sourcing from regional scale products [*Brennan et al.*, 2016], our model configuration and domain best benefit from our

chosen pairing of a global product with regional climatology correction. The Oregon State University Tidal Prediction Software

(OTPS) [*Egbert and Erofeeva*, 2002] was used to develop harmonic tidal forcing of sea level and depth-average velocity along

the open boundaries.


In order for the model sea level to be of use for further downscaling applications, such as driving sea level boundary conditions

to higher resolution estuarine models, or studies of coastal inundation, we needed to adjust our mean dynamic topography during

the bias correction to a useful local reference a datum; here, the NAVD88 (North American Vertical Datum of 1988). Coastal sea

level from a preliminary simulation was compared to 14 NOAA tide gauges in the MAB and GOM that report sea level with

respect to NAVD88. These showed an almost uniform bias offset of 0.1959 m, so this adjustment was made to the MDT derived

from the climatological analysis. This has no dynamical impact, but effectively reconciles the global and regional datums so that

sea level output from Doppio is a best estimate of the total water level at the coast with respect to the regionally accepted datum.

**3 Skill Assessment**

The model performance was assessed by comparison to a comprehensive aggregation of all available observational data for

2007—2017 from in situ and remotely sensing platforms. The aggregation includes sea surface temperature (SST) from several

infrared and microwave sensing satellites, sea surface height (SSH) from satellite altimeters and in situ tide gauges, velocity from

surface-current measuring HF-radar, and in situ temperature and salinity from numerous operational and research platforms. A

full list of data types and sources is presented in Table 2, and the total number of observations per month for each source of

observational data is shown in Fig. 5. A ROMS option (VERIFICATION in Table 1) activates extracting model values at points

in time and space by bi-linear interpolation to match the coordinates of data in observation files formatted for ROMS 4-D

Variational (4D-Var) data assimilation. At run time the VERIFICATION option creates a separate output file populated with

interpolated model values corresponding to all observations, enabling various statistical comparisons. These observation files are

the end result after quality control screening of the raw data streams, and the creation of binned averages of sources where the

resolution exceeds that of the model (notably SST and dense in situ trajectory-profile data from gliders), and decimation of

CODAR velocity to give independent observations, consistent with standard practice in the ROMS 4D-VAR framework.

[Table 2]

[Figure 5]

For the statistical skill assessment analysis, five sub-regions of the Doppio domain representing distinct dynamical regimes were

considered – anticipating that model performance may exhibit varying skill in different situations. These are the Scotian Shelf,

the Gulf of Maine, Georges Bank, the Mid-Atlantic Bight, and the Shelf break to 3500 m (Fig. 6). Aggregated model-observation





difference statistics within each region, independent of any further spatial or temporal distinctions, are shown for temperature (both SST and subsurface), salinity and vector velocity model skill in Fig. 6. The results are evaluated in the form of Taylor diagrams [*Taylor*, 2001], a robust method of visualizing multiple statistical parameters within a single plot. Figure 6 includes an explanatory schematic Taylor diagram. Normalizations are with respect to the observation standard deviation. The normalized Centered Root Mean Square (CRMS) error is indicated by dashed arcs that show proximity to the unfilled marker on the x-axis

at (1,0). The normalized standard deviation is shown as distance from the origin (0,0) indicated by dotted lines, with the unit arc indicating the model and observation standard deviation match. Along the outer curved edge is shown the correlation coefficient. Lastly, the normalized mean bias is shown as the stick originating from each marker, where the distance from the tip to the aforementioned unfilled marker along the x-axis is the normalized uncentered RMS error.

[Figure 6]

During our model configuration, design and testing, various options were evaluated using the VERIFICATION framework to systematically determine if they led to quantitative improvement in model skill. These experiences are instructive for the design of ROMS -based regional ocean models in general, so we briefly outline results for these tests below to complete the description

of the Doppio configuration.

### 3.1 Surface Stress from Wind Relative to Surface Current

A feature of the Doppio configuration that differs from widespread ROMS practice is a modification to the bulk formula to use wind velocity relative to the surface current in the calculation of surface stress [*Bye and Wolff*, 1999]. Typically, ROMS models

do not make this correction though it has been an option for some time. In a Taylor diagram for sea surface current skill (Fig. 7(a)) we see a modest but consistent improvement in model skill when incorporating the wind-current difference in the stress calculation that warrants its incorporation in the standard Doppio configuration (WIND_MINUS_CURRENT in Table 1). Scotian Shelf markers are absent from Fig. 7(a) because most of the velocity observations are from CODAR, which are predominantly for the MAB, with some coverage extending slightly into the Shelf break to 3500 m and to Georges Bank. Few

velocity observations from NERACOOS (Northeastern Regional Association of Coastal Ocean Observing Systems) moorings in the Gulf of Maine are close to the surface and they are not instructive in evaluating the bulk formula parameterization of stress.

[Figure 7]

### 3.2 Precipitation Forcing

The NARR precipitation analysis over the ocean assimilates satellite derived rainfall data from the Climate Prediction Center (CPC) Merged Analysis of Precipitation (CMAP) [*Xie et al.*, 2007] south of 20° N but gradually transitions to no assimilation at all north of 50° N. This raises some uncertainty as to the validity of the precipitation forcing for Doppio, so we have evaluated substituting NARR precipitation values entirely with data from NASA's TRMM (Tropical Rainfall Measuring Mission) Multi-

satellite Precipitation Analysis (TMPA) [*Huffman et al.*, 2007]. In Fig. 7(b), we can see that Doppio with TMPA precipitation





forcing (square marker) results in very comparable model skill to NARR (circle marker) for salinity for most of the domain, although it is marginally worse in the MAB and Gulf of Maine. Therefore, we opted to keep the NARR forcing over TMPA.

### 3.3 Open Boundary Bias

The Doppio open boundary conditions are taken from the Mercator-Océan product, with an annual mean bias correction applied to bring it into agreement with our MOCHA regional climatology. To illustrate the improvement this makes, Fig. 7(c) contrasts the model skill for SST, in situ temperature, and salinity when running with the uncorrected Mercator-Océan (filled symbols) and the bias corrected version (unfilled symbols). The decrease in the bias vector is to be expected. But it is also evident that bias correction carries with it modest improvement in correlation across the domain.


While we use Mercator-Océan products for boundary data, a popular choice by other users for regional models is the HYCOM (HYbrid Coordinate Ocean Model) global model analysis product by the Global Ocean Data Assimilation Experiment (GODAE) [*Chassignet et al.*, 2009; *Metzger et al.*, 2014]. In Fig. 7(d) we compare model skill for SST and in situ temperature, and salinity when using the HYCOM open boundary data without bias correction (filled symbols) to that of the bias corrected Mercator files 305 in Doppio (unfilled symbols). The model skill using HYCOM is inferior to the case using Mercator open boundary data.

### 3.4 Velocity and Sea Level Coherences

NERACOOS mooring data are valuable for skill assessment in the comparison of model and observed velocity time series in the form of frequency domain coherences (Fig. 8) for the three long-term velocity time series at sites B, M and N. The spectra are computed by standard periodogram smoothing [*Moore and Wilkin*, 1998] with red lines showing 90 % confidence. The model 310 has intrinsic skill in coherence at all time scales in the coastal current (site B). At the Northeast Channel entrance to the GOM (site N) the model captures high frequency and seasonal timescales, but falters in the mesoscale. This suggests model performance may well improve with the assimilation of mesoscale resolving observations of sea level from satellite altimetry. In the central GOM (site M) coherence is only significant on timescales shorter than 20 days, presumably in response to well modeled local forcing. At this site, also, the assimilation of mesoscale resolving data could improve simulation of intermediate 315 time scales that impact stirring and mixing in the GOM.

[Figure 8]

In Sect. 2.3, data from 14 NOAA tide gauges were introduced in referencing mean model sea level to the regional NAVD88 320 datum. In Fig. 9 we present frequency domain coherence for 6 representative sites (see inset map) distributed across the Mid-Atlantic Bight and the Gulf of Maine. Sea level variability is statistically significantly coherent across all resolved scales throughout the region.

[Figure 9]



## 4 Results and Discussion

The seasonal cycles of sea surface (red) and bottom (blue) temperatures from the model are shown in Fig. 10, with interannual variability depicted in the shaded envelope and the 11-year mean indicated by the thicker lines. In the winter months, the temperature in all shelf regions at the sea surface drops below the temperature at the seafloor. The increase in seasonal bottom temperature lags behind sea surface temperature, with typically 2 to 3 months passing after peak summer temperatures before the bottom cooling that marks the breakdown of stratification and deeper mixing of the thermocline; this is most evident in the Gulf

of Maine and Mid-Atlantic Bight. The lack of variability in the bottom temperature for the shelf-break to 3500 m region is expected given the order of magnitude difference in depth compared to the other regions along the shelf.

[Figure 10]

Model skill in capturing the seasonal cycle of vertical stratification is presented from a different perspective in Fig. 11, showing ensemble mean vertical profiles (upper 250 m only) of temperature and salinity for four representative months in the various sub-regions. The variability in model (solid line) and the observations (dashed line) show similar behavior. The comparison is made by interpolating the model to available data coordinates as in Sect. 3, and binned at 10 m vertical intervals above 100 m depth, and 50 m intervals below that. The vertical extent of the comparison varies through the year depending on data availability. The

model shelf waters have a tendency to be slightly cooler than the observations below 100 m, while also being warmer at the surface during the summer months and cooler at the surface during the winter months. The cooler model temperatures during September in Fig. 11 could be due to the bias correction applied along the boundaries; as the correction uses a harmonic analysis, the fall overturning circulation could result in too much variability resulting in those cooler model temperatures. The seasonal cycle of salinity stratification is modeled well in shelf waters, with a tendency to slightly high salinity in the range 25 m to 100 m

during the summer months, most notably on the Scotian Shelf. The model-observation difference is generally less near both the seafloor and sea surface. A characteristic pattern throughout the region is the elevated salinity with depth that maintains water column stability in the face of the weak seasonal thermal inversion.

[Figure 11]


    Ocean temperature at the seafloor is a strong driver of shellfish and fisheries ecology throughout the MAB and GOM [*Drinkwater et al.*, 2003; *Murawski*, 1993; *Sullivan et al.*, 2005], so to evaluate the model's ability in this regard a unique set of observations were used for the comparisons in Fig. 12, these being fishing trawler-collected bottom temperatures acquired through a project coordinated by the Northeast Fisheries Science Center (NEFSC) [*NMFS*, 2015] . The left column of Fig. 12

shows 2-dimensional histograms of bottom temperature observed by sensors mounted on fishing vessel trawl doors versus Doppio. The right column shows the corresponding geographic spread of the observations colored by the difference in model minus observed bottom temperature. The rows of Fig. 12 group the comparisons by the same four months used in Fig. 11. For the purposes of this analysis, we consider observations reported to be less than one tenth of the water depth above the model seafloor to be "bottom" observations. The histograms show overall good correspondence between model and the fishing fleet

observations with any bias being generally small. Model skill is consistently good throughout the GOM and on Georges Bank. In early Spring (March) there is a tendency toward a model cool bias along the shelf-break, but with a dense cluster of observations





near 71° W from the New Bedford fleet showing an opposing warm bias. There is an opposite sense to the model bias in this area in winter (December) with the shelf-break slightly warm and mid-shelf slightly cool. The standard deviation of model-data discrepancies is not especially low (typically ~4° C), but we have not attempted to aggressively quality control this data set with

respect to the number of independent samples that enter each reported observation, or the depth variance of samples in each aggregate observation. Such an effort will be required before these data are adopted in a data assimilation system.

[Figure 12]

Returning to statistical evaluations of model-data differences using Taylor diagrams, we wish to delve further into regional differences and make some comments on the accuracy of some of the data sources used in our skill assessment approach.

The Doppio model temperature skill compared against observations is presented separately for different satellite sea surface temperature products and in situ observing platforms in Fig. 13(a). Upward pointing triangle symbols are for SST; downward for

in situ temperature. For most observing networks there is good statistical agreement between Doppio temperatures both at the surface and at depth. Clustering at the unit radius indicates the model variance is close to observed, with strong correlations in the vicinity of 0.9. Bias is small. Of interest is that there is a clear discrepancy for the TRMM Microwave Imager (TMI) SST data against the other satellite products. The TMI sensor only provides data south of 38° N and more than 100 km from the coast, so the statistics are skewed strongly toward model results in the Slope Sea and Gulf Stream. Nevertheless, that the skill should be

so dramatically different in comparison to other microwave SST sensors (WSAT and AMSR) is troubling. WSAT and AMSR have comparable spatial coverage and resolution, yet skill for WSAT is significantly poorer than for AMSR. While we have retained these data in the sub-region temperature analysis (Fig. 13(b)) we suspect that TMI and WSAT may not be reliable in the Doppio domain, and will withdraw them from future data assimilative reanalyses. The in situ temperature comparisons show strong agreement that is as good as infrared SST sensors, so we are confident in Doppio's ability to simulate temperature not

merely at the surface but throughout the water column.

In Fig. 13(b) we separate the evaluation according to our standard sub-regions, and again contrast surface (satellite) and sub-surface (in situ) with directed triangle symbols. Model performance is best in the GOM and over Georges Bank, though with a bias over Georges Bank that stems from the September and December results already noted in Fig. 11. In comparing SST and

sub-surface temperature for the Scotian Shelf and Mid-Atlantic Bight, we see that the Doppio model does well for SST but less so for sub-surface temperature. This is perhaps unsurprising given the strong constraint that prescribed meteorological conditions exert on ocean circulation model SST.

[Figure 13]


In assessing Doppio's skill in simulating mixed layer depth (MLD) variability within the Gulf of Maine and over Georges Bank, we compare (Fig. 14) modeled to observed frequency of occurrence of mixed layer depths [*Christensen and Pringle*, 2012] using a common MLD definition: the depth where the potential density is 0.01 kg/m$^3$ greater than at the sea surface [*Thomson and Fine*, 2003]. We note that Doppio best simulates hibernal mixed layers along the Gulf of Maine's coast, and tends to have a

slightly deeper than observed mixed layer in the other zones of the Gulf of Maine and Georges Bank. It is worth noting that the



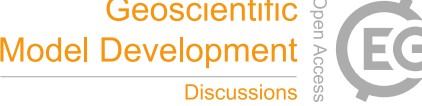

coast zone has the best spatial coverage in sampling of all zones, whereas in other zones the coverage is not nearly as uniform. The model estimated MLD is a uniform sub-region average and may not sample the ocean equivalently to *Christensen and Pringle's* [2012] analysis where their sample size was small.

[Figure 14]

Figure 15(a) shows the 5-year (2007-2012) average Mean Dynamic Topography for the free running Doppio model, which is constrained at the perimeter by bias corrected open boundary data corresponding to Fig. 15(b) – the outcome of the 4D-Var climatology analysis following the *Levin et al.* [2018] methodology, which we consider the best representation of regional MDT.

The differences between the two would likely diminish with assimilation, but the free running Doppio still well represents the coastal waters, especially the coastal current of the GOM. Also shown (Fig. 15(c)) is the MDT from Mercator product PSY4QV3R1, from which Doppio's open boundary conditions were adapted. Of note is the inaccurate GOM circulation, which is understandable due to the regional-scale tradeoffs a global model must make, especially in the GOM where tidal dynamics (which are absent from Mercator) are such an important driver of mixing and circulation. Figure 15(d) shows MDT from AVISO

product CNES-CLS13, on which the aforementioned Mercator product is based. Both Mercator and AVISO show a preponderance of dynamic height contours intersecting the coast, which would imply surface geostrophic currents normal to the shore. While both the AVISO and Mercator products have been superseded, and much improved in their regional definition, since 2013 when the Doppio system was being created, it was the lack of physically reasonable GOM circulation structure that prompted our independent pursuit of a kinematically and dynamically balanced regional MDT [*Levin et al.* 2018]. In our

judgment, the free running Doppio MDT is still more accurate than the AVISO CNES-CLS18 product (not shown) and our 4D-Var climatology analysis remains the most accurate regional portrayal of the system.

[Figure 15]

Shown in Fig. 16 are the mean model circulations from the same 5-year (2007-2012) period as in Fig. 15, for the upper 100 m and for below, overlaid upon the bathymetry binned to emphasize isobaths. Evident from these are the Gulf of Maine's main oceanic inflows: Scotian Shelf water coming along the Halifax coastline and originating from the Labrador Current; and Slope Sea water entering through the Northeast Channel that derives from subpolar North Atlantic mixed with eddies of the Gulf Stream. The two main outflows are water exiting near Nantucket, and waters exiting out the Northeast Channel and around

Georges Bank in alignment with the accepted general circulation pattern [*Brown and Beardsley*, 1978]. Circulation within the deep basins of the GOM is also evident, and the GOM coastal current is pronounced at the surface. The general southwestward flow on the MAB shelf, modified by an offshore Ekman component, is clear.

[Figure 16]






## 5 Summary

This article has described in detail the features of a ROMS-based regional circulation model, Doppio, for the Mid-Atlantic Bight and the Gulf of Maine. The model downscales open boundary information drawn from Mercator-Océan or global HYCOM, but we have shown that taking steps to adjust for biases in these global class models leads to discernable improvements in Doppio

performance. The model demonstrates useful skill in comparison to a comprehensive suite of satellite and in situ observations from a dense coastal regional integrated ocean observing network. There are aspects of the model solution that would likely improve with formal data assimilation, but that is not part of this body of work. The configuration uses surface, river and open boundary forcing data streams that are suited to real-time operation, and such a system with 4-dimensional variational (4D-VAR) data assimilation [*Levin et al.*, 2019; *Wilkin et al.*, 2018] has been prototyped for MARACOOS.


The focus of development was on achieving a model configuration that allows for decadal scale simulations of physical ocean circulation that can ultimately underpin regional studies of ecosystems and biogeochemistry. As such, faithfulness to stratification throughout the entire water column, especially in coastal and shelf waters, is paramount. Doppio captures both the temperature and salinity stratification well, including a region-wide vertical salinity gradient that maintains stable water columns

in the face of winter temperature inversions. Bottom temperature is a particularly challenging aspect to model in this region because of the extreme vertical stratification that arises in summer in the MAB, and the influence of warm offshore waters at the shelf edge that are the principal driver of seasonal temperature inversions on the outer shelf. To affirm the model performance in regions relevant to ecosystems and fisheries, comparison was made to near seafloor temperature data acquired from trawl fishing gear with encouraging results. A further aspect of regional dynamics relevant to ecosystems is mixed layer depth, and where

reliable climatological analyses exist, in the Gulf of Maine coastal current, the model performance in acceptable. The large scale mean circulation of the region as characterized by Mean Dynamic Topography from data assimilative climatological analyses is preserved in the Doppio simulations.

It is anticipated that this Doppio model set-up will see similarly broad adoption for studies of ecosystems, biogeochemical cycles

and ocean weather, as was the case for the ESPreSSO system as noted in Sect. 1. Decade-long simulations are already in progress to examine transport pathways and timescales of open ocean to shelf and marginal sea exchange, and coupled physical-biogeochemical interactions. Future developments are also underway to enable higher spatial resolution (~ 2 km) that admits submesoscale variability, which may in turn have significant impact on modeled ecosystems.

*Code & Data Availability*

Doppio uses the ROMS source code (ROMS version 3.6, SVN revision 898) with specific configuration options detailed in Table 1. The code is accessible for free download at the ROMS website (http://www.myroms.org), and is open-source licensed according to the MIT/X license (opensource.org/licenses/mit-license.php). Doppio forcing, boundary and other user input files are available on request via the Rutgers Ocean Modeling Group (OMG) THREDDS (Thematic Real-time Environmental

Distributed Data Services) Data Server (TDS). The 2007-2017 Doppio model output is also accessible via OMG TDS. Validation data and model-data differences from the VERIFICATON analysis are available via the OMG ERDDAP service.





*Author Contributions*

Much of the original model design was done by JW and JL, with AL creating select forcings. JW and JL collected and
maintained observational data aggregation used for model verification. AL conducted configuration experiments that guided
model development, as seen in the earlier skill assessment. Model results interpreted and evaluated by AL with guidance by JW
and JL. All authors contributed to the paper.

*Competing Interests*

The authors declare that they have no conflict of interest.

*Acknowledgements*

The research and development of Doppio has been funded by NOAA grants NA11NOS0120038 and NA16NOS0120020, and
indirectly via other support by NASA, NOAA, NSF, and ONR. The authors thank the international ROMS user community on
www.myroms.org for the continued discussion and advancement of the ROMS project.

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



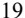

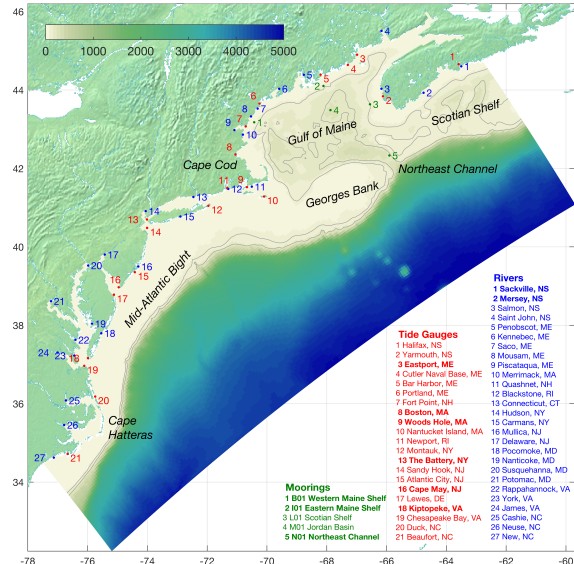

**Figure 1. Doppio bathymetry with markers for all rivers used to force the model, and tide gauges and moorings used for statistical comparisons. Those in bold are referenced in Figs. 4, 5, 8, 9 and 13.**

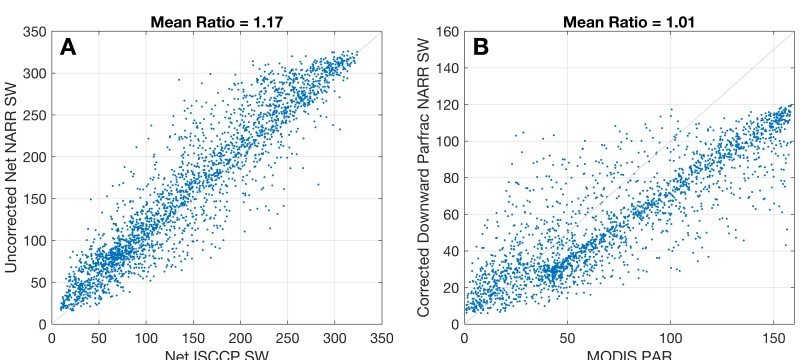

**Figure 2. (A)** Net NARR shortwave daily averaged and uncorrected for overestimation compared to ISCCP net shortwave; mean ratio indicates 17% over-estimation. **(B)** NARR shortwave with 23% reduction for over-estimation [*Wang et al.* 2012], 6% reduction for albedo [*Payne*, 1972] and assume fraction of PAR of 0.45 [*Kirk*, 2010] compared to MODIS satellite PAR.

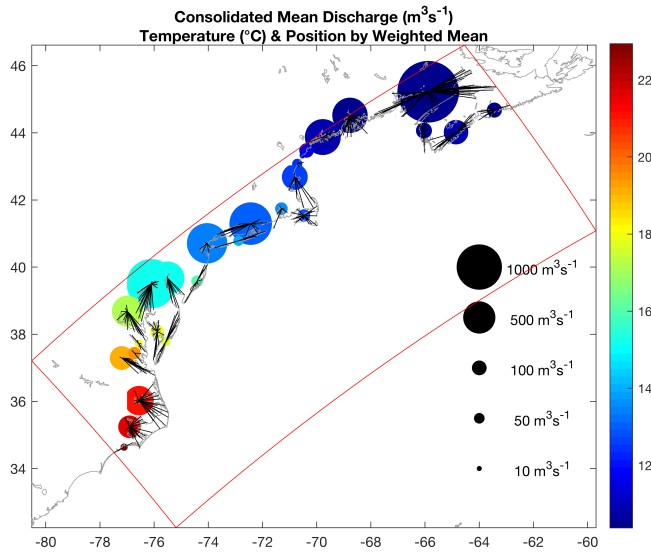

**Figure 3. River model aggregations. Point size indicates discharge volume; color indicates mean temperature in Celsius. Line segments link individual rivers to their discharge-weighted mean location and illustrate effective watershed extent. Red perimeter denotes Doppio domain boundary.**

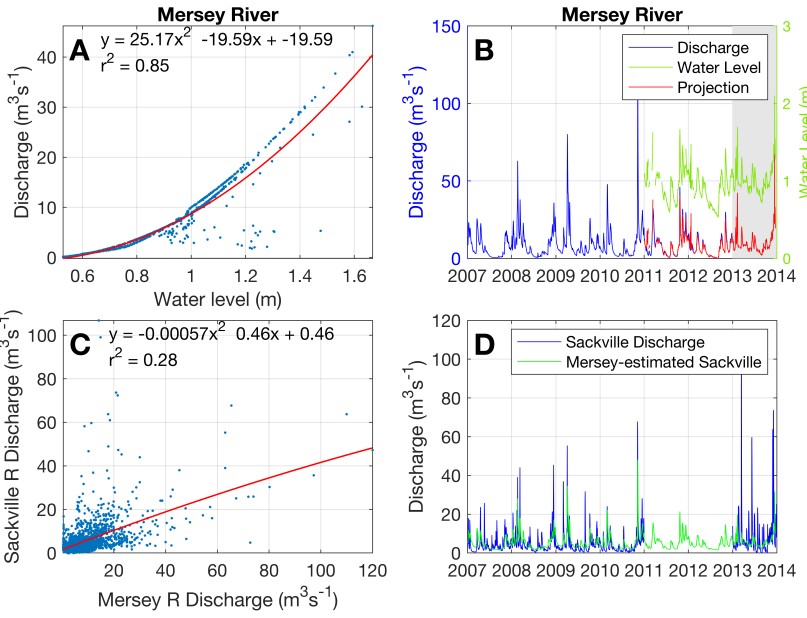

**Figure 4. (A)** Discharge to water level relation for Mersey river for the reanalysis period. **(B)** Projection of discharge via water level (using relation in (A)) for periods of missing discharge data indicated by gray background). **(C)** Comparison between Mersey river and Sackville River discharge over the reanalysis period to find a suitable relation. **(D)** Observed Sackville discharge (blue) and predicted Sackville discharge (green) based on relation in (C). Mersey and Sackville River locations are noted in Fig. 1.



## Total Number of Observations per Month for 2007-2017



Figure 5. Total observations by month for the individual data sources used in skill assessment over the 2007-2017 period. The log color scale indicates quantity of observations.

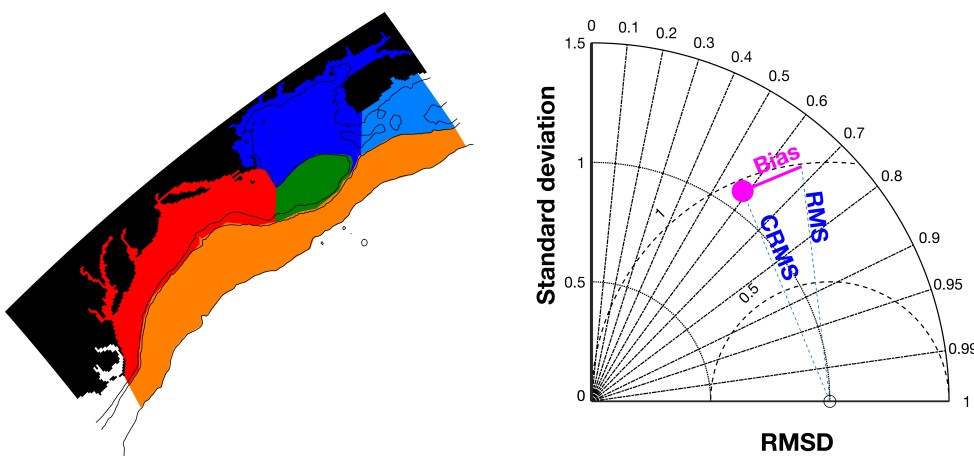

**Figure 6. Left: The 5 model domain sub-regions used to better distinguish geographic variation in skill performance. Right: Schematic Taylor diagram. Radial distance is model standard deviation normalized by observation standard deviation; azimuth is the arc cosine of the correlation; distance to point (1,0) on the x-axis is the normalized centered RMS error. Stick indicates normalized mean bias of the model; distance from the end of the stick to (1,0) is the overall normalized RMS error including bias. The closer to (1,0), the better the performance.**

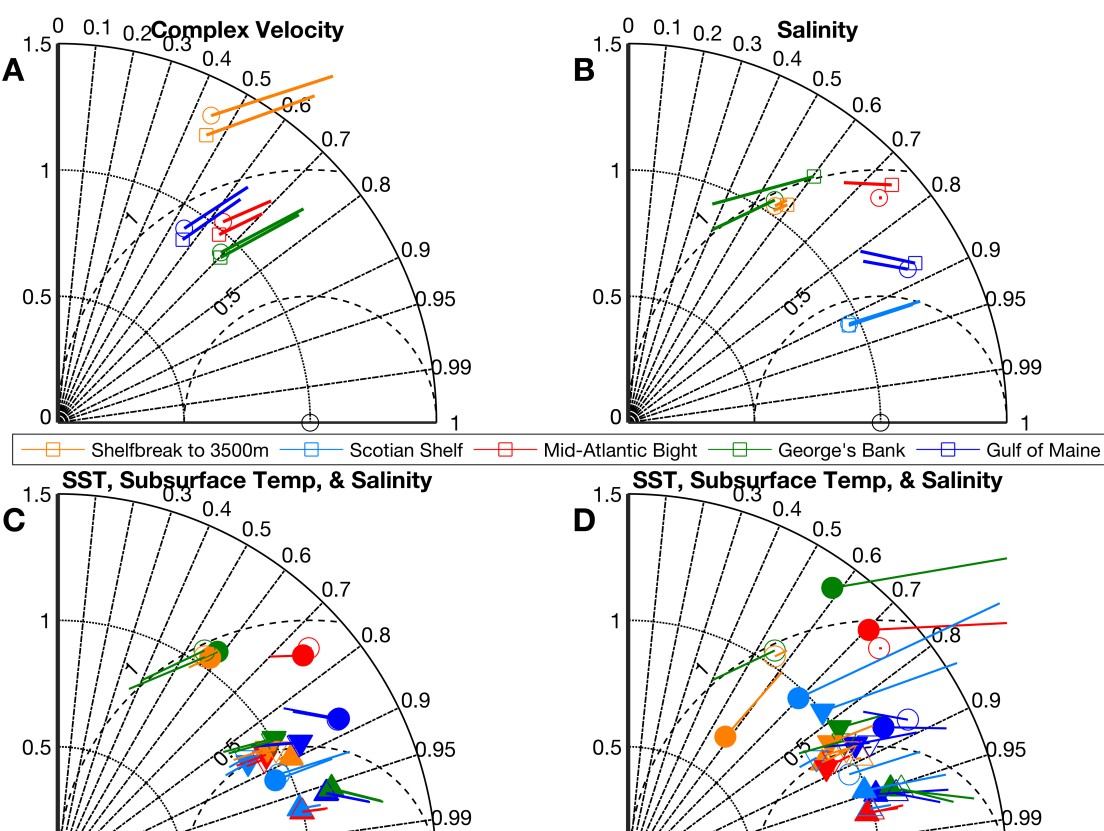

**Figure 7.** Taylor diagrams of model skill for the different model setup cases. Symbols are colored according to sub-regions defined in Fig. 6. (A) Complex velocity, shown with (square) and without (circle) the change to the bulk formula calculation for wind stress relative to surface current; (B) Salinity, shown for TMPA (square) against NARR (circle) precipitation forcing; (C) SST (Δ), subsurface temperature (∇), and salinity (circle), shown for Mercator open boundary data not corrected for bias (filled) against bias-corrected open boundary Mercator data (unfilled); (D) SST (Δ), subsurface temperature (∇), and salinity (circle), shown for HYCOM open boundary data not corrected for bias (filled) against bias-corrected open boundary Mercator data (unfilled).

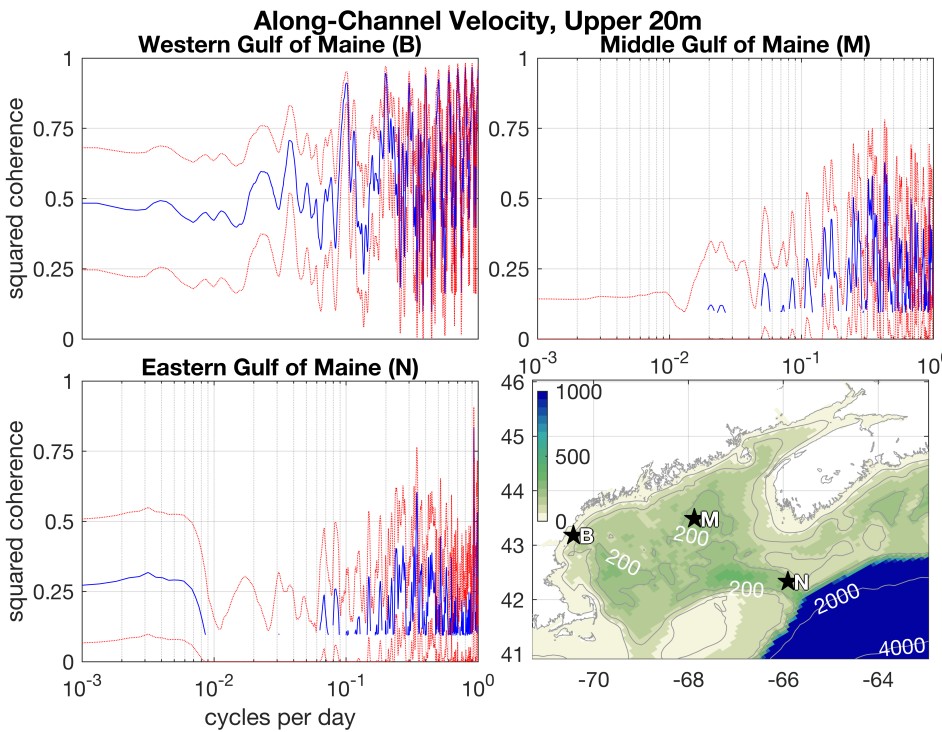

**Figure 8. Velocity coherences (blue) with error bars (red) for three representative moorings across the Gulf of Maine. Doppio has intrinsic skill in coherence at high frequency and seasonal timescales, but falters in mesoscale. The coastal current variability is captured well at all time scales. These mooring locations are also noted relative to the whole domain in Fig. 1.**





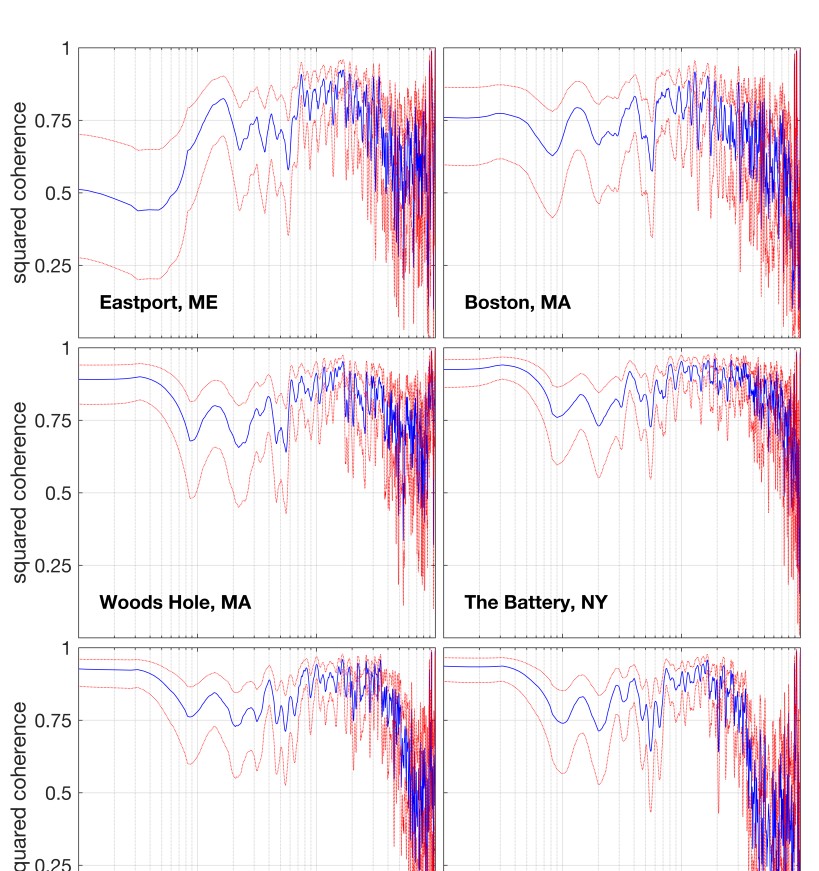

**Figure 9. Sea Level Coherence (in blue, error bars in red) along the domain's coastline from tide gauge data against model output. Tide gauge locations are noted in bold in Fig. 1.**

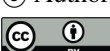

Figure 10. Ensemble seasonal cycles for sub-regions defined in Fig. 6 for 2007–2017 simulation. Red is sea surface temperature; blue the bottom temperature. Thick line is 11-year mean; thin lines represent individual years and the shaded envelope shows the spread of interannual variability.



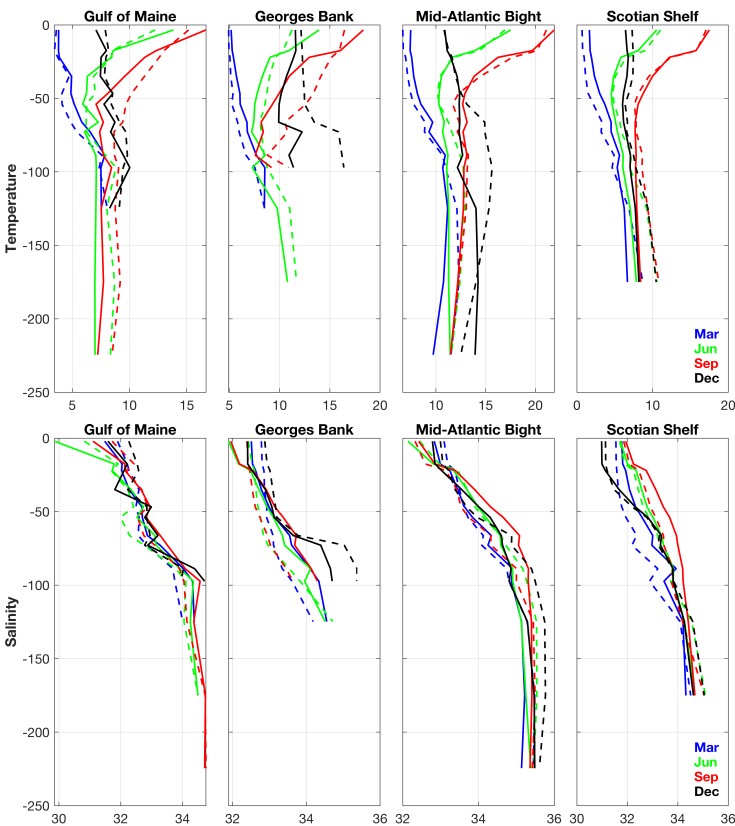

**Figure 11. Vertical profiles of temperature and salinity, binned every 10 m for the first 100 m, then binned every 50 m below 100 m, for four representative months, for the upper 250 m. Solid lines are the model profiles; dashed lines are the observation product profiles. Top row: Temperature. Bottom row: Salinity.**



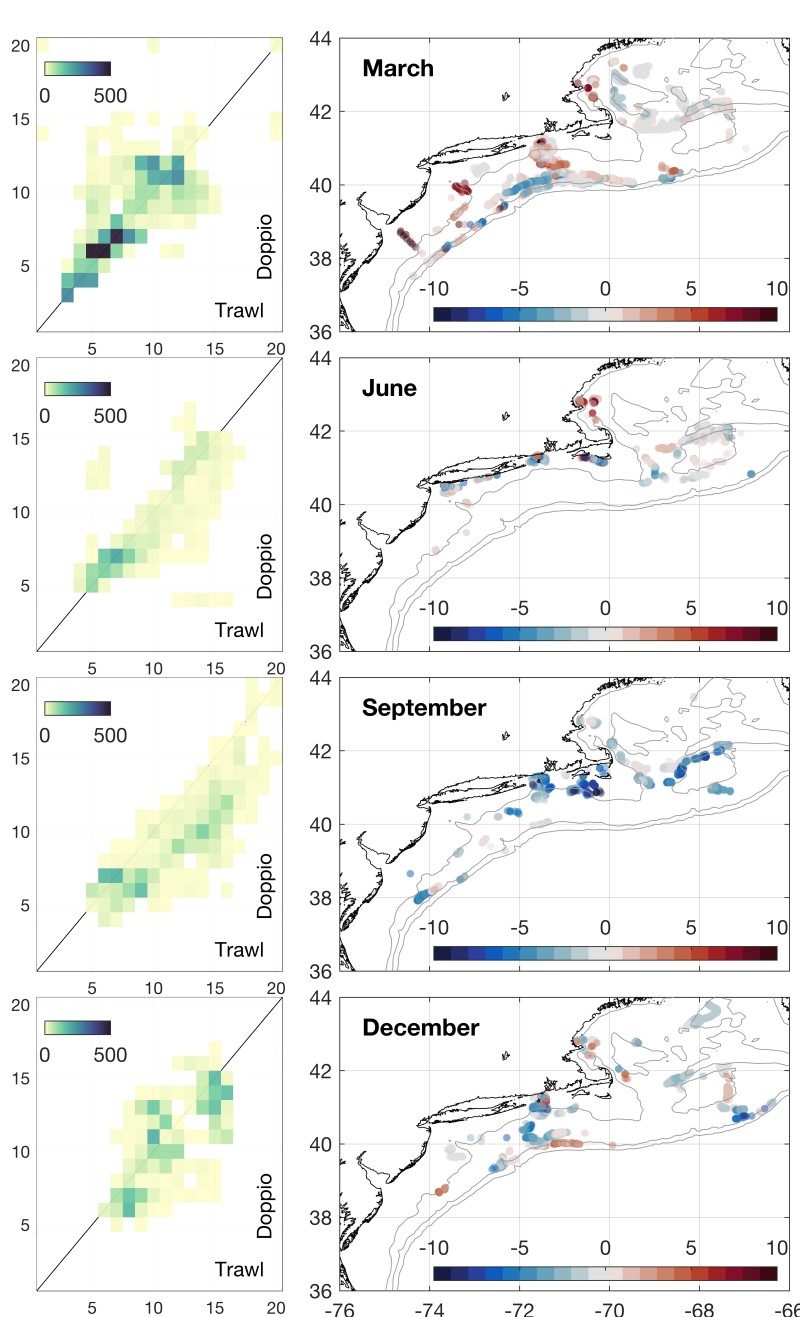

**Figure 12. Comparison of bottom temperature observations from fishing trawl data to Doppio for data collected from 2007–2017. Left: 2-D histograms (°C). Right: Positions of model-data match-up comparisons, colored by the temperature difference Doppio minus observed (°C) where red (blue) means the model is warmer (cooler) than observations.**

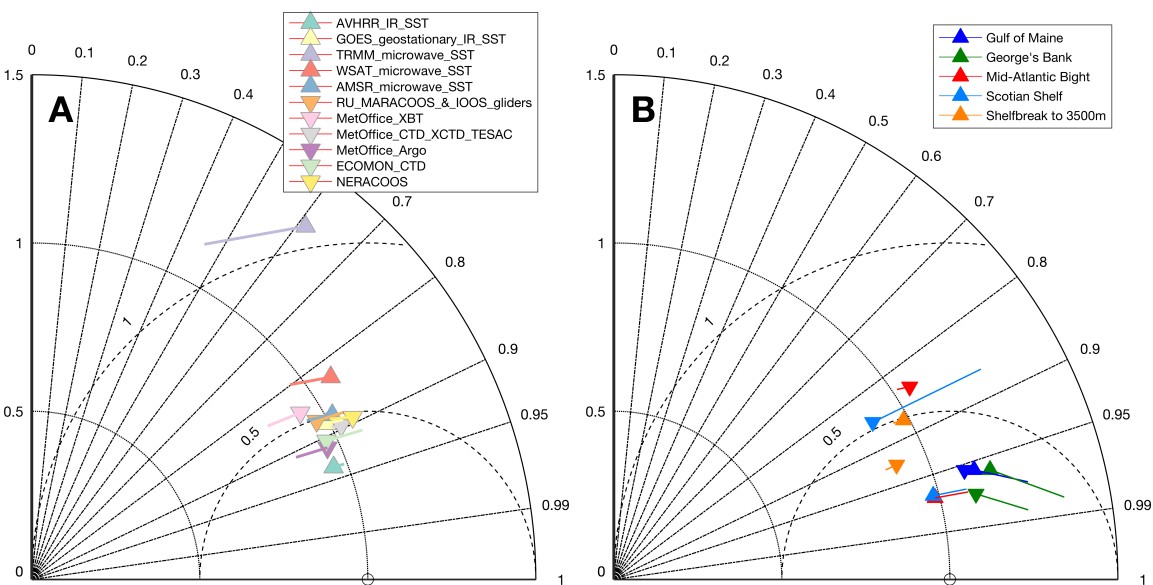

**Figure 13. Taylor diagrams. (A)** Doppio model skill against temperatures from satellite products including AVHRR, GOES, TRMM, WSAT, and AMSR, and in situ observations throughout the water column, including gliders from MARACOOS and IOOS, UK MetOffice XBTs, CTDs, Argo floats, ECOMON CTDs and NERACOOS moorings. **(B)** Doppio model skill against the suite of observations represented in (A) but split into the five sub-regions (see Fig. 6) with Δ for sea surface temperature and ∇ for subsurface temperature.



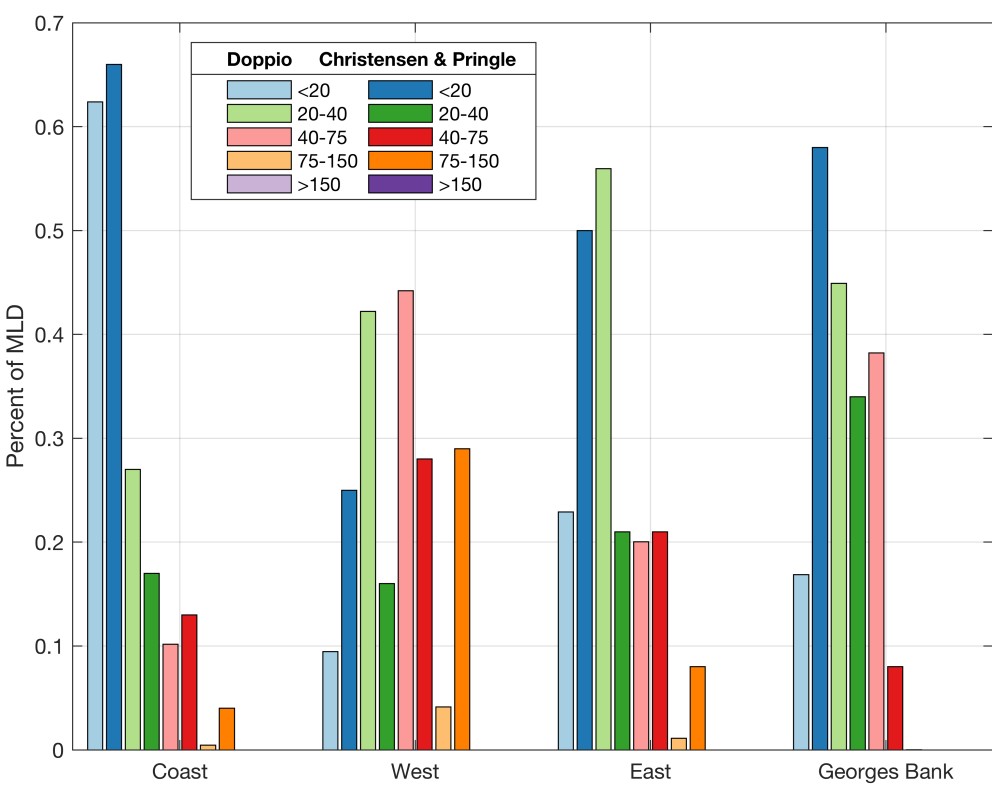

**Figure 14. Comparison of frequency of occurrence of mixed layer depths (modeled and observed) for zones in the Gulf of Maine, and Georges Bank. MLD definition and specific zones follow *Christensen and Pringle* [2012].**

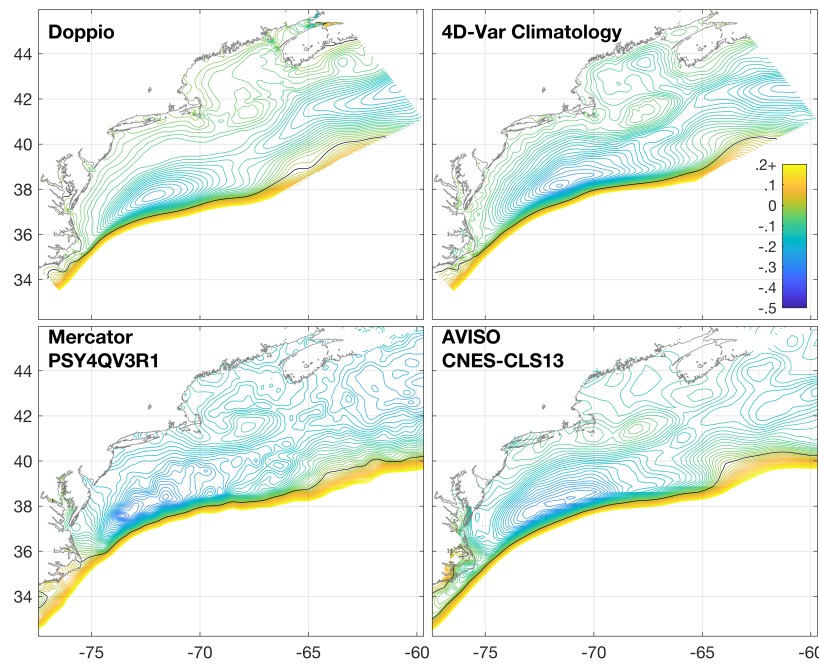

**Figure 15. Mean Dynamic Topography (MDT) (meters) in the model domain. (A) Five-year (2007-2012) mean of Doppio. (B) Our 4D-Var climatology analysis. (C) Mercator product. (D) Global AVISO product.**



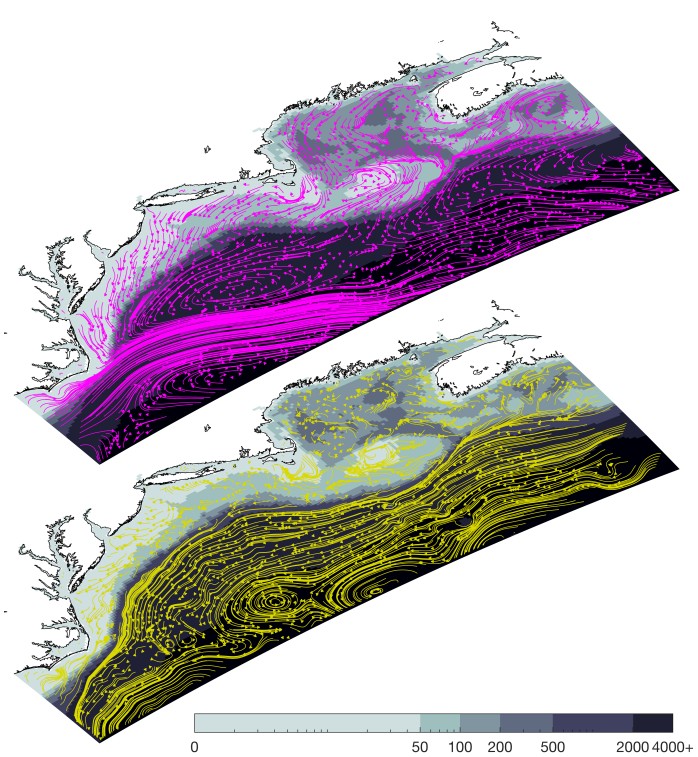

**Figure 16. Five-year (2007-2012) mean model circulation overlaid on bathymetry. Vector resolution has been decimated by 10 on the shelf and 20 beyond the shelf. Top: Mean circulation for the top 100 meters (magenta). Bottom: From 100 meters to the seafloor (yellow). The vectors are extended to show transport displacement for durations of 10 and 50 days, respectively, in the top and bottom panels.**



| Configuration | CPP / Logical Flag Option | Details | Parameter | Value |
|---|---|---|---|---|
| Horizontal Advection | TS_A4HADVECTION UV_U3HADVECTION | 4th order Akima for tracers, 3rd order upstream for momentum | DT | 360.0 s |
| Vertical Advection | TS_A4VADVECTION UV_C4VADVECTION | 4th order Akima for tracers, 3rd order centered vertical for momentum | NDTFAST | 30 |
| Horizontal mixing of momentum | MIX_S_UV UV_VIS2 | Harmonic mixing on s-coordinate surfaces | nl_visc2 | 100 m²s⁻¹ |
| Horizontal mixing of tracers | MIX_GEO_TS TS_DIF2 | Harmonic mixing on z-coordinate surfaces | nl_tnu2 | 20 m²s⁻¹ |
| Vertical turbulence closure | GLS_MIXING, CRAIG_BANNER, KANTHA_CLAYSON, N2S2_HORAVG, SPLINES_VDIFF, SPLINES_VVISC | Generic length scale $k$-$kl$, Craig & Banner surface flux, Kantha & Clayson stability function, H smoothing of buoyancy/shear, Splines reconstruction for V shear | | |
| Pressure gradient | DJ_GRADPS, ATM_PRESS | Splines density Jacobian (Shchepetkin, 2000), Inverse Barometer | | |
| Open boundary conditions momentum | LBC = Cha (free surface), Fla (2D), RadNud (3D), SSH_TIDES, UV_TIDES,ADD_FSOBC, ADD_M2OBC | 2-D Chapman implicit and Flather with OTPS harmonic tides (MS4, M4, MN4, K2, S2, M2, N2, K1, P1, O1, Q1). 3-D radiation with nudging to data assimilative climatological analysis | | |
| Open boundary conditions tracer | LBC = Rad | Radiation with nudging to regional climatology | | |
| Surface momentum flux | BULK_FLUXES, WIND_MINUS_CURRENT | Fairall bulk fluxes, Wind forcing on sea surface current, outgoing | | |
| Surface heat flux | LONGWAVE_OUT, DIURNAL_SRFLUX | longwave radiation, local diurnal cycle for shortwave radiation, NARR & NAM meteorological data | | |
| Surface salt flux | | | | |
| Vertical penetration of solar shortwave radiation | SOLAR_SOURCE | | WTYPE | 4 |
| Bottom drag | UV_QDRAG | Quadratic Drag | rdrg2 | 0.003 |
| River Sources | | Gauge data from United States Geological Survey & Water Survey of Canada | LuvSrc | T |

Grid Resolution: 7 km horizontal, 40 level vertical s-coordinate Vtransform = 2, Vstretching = 4, $\theta S$ = 7, $\theta B$ = 2, hC = 250 m.

Source code version: svn revision 898

**Table 1. Key C-preprocessing options and ROMS parameters chosen for the Doppio model configuration.**



| Observation type and platform | Sampling frequency and resolution | Source |
|---|---|---|
| AVHRR infrared SST (Sea Surface Temperature) | 4 passes per day, super-obs at 7 km | MARACOOS.org and NOAA Coastwatch |
| GOES infrared SST | 3-hourly, 6 km | NOAA Coastwatch |
| AMSR and WindSat microwave SST | daily, 15 km | NASA JPL PODAAC |
| In situ temperature and salinity (T, S) from NDBC buoys, Argo floats, shipboard XBT and surface drifters reported to Global Telecommunication s System (GTS) | varies with platform | NOAA Observing System Monitoring Center (OSMC) |
| In situ T, S from IOOS autonomous underwater glider vehicles | ~1-2 deployments per month, dense along trajectory | U.S. Integrated Ocean Observing System (IOOS) Glider DAC |
| In situ T, S from CTD casts of NOAA Ecosystem Monitoring voyages | 2 surveys per year, ~24,000 data points | NOAA Northeast Fisheries Science Center (NEFSC) |
| In situ T, S from quality controlled historical data set archive | varies | UK Met Office |
| In situ T from sensors mounted on lobster traps and trawler fishing gear | Varies, ~76,000 data points | NOAA NEFSC e-Molt program |
| Surface currents from CODAR HF-radar | hourly, 7 km | MARACOOS.org THREDDS Data Server |
| Satellite altimeter SSH (Sea Surface Height) from Envisat, Jason series, AltiKa and CryoSat | ~1 pass each day within model domain, ~7 km | Radar Altimeter Database System (RADS) at TU Delft |
| Tide gauges from NOAA CO-OPS | Hourly, 21 gauges | Tides and Currents NOAA |

Table 2: Observations used in model-data skill assessment, their nominal resolution, and origin.