# Peer review of "Doppio – A ROMS(v3.6)-based Circulation Model for the Mid-Atlantic Bight and Gulf of Maine: Configuration and comparison to integrated coastal observing network observations"

_Geoscientific Model Development, 2019_

## Short Comment (SC1) · 26 Mar 2020

Dear authors,

in my role as Executive editor of GMD, I would like to bring to your attention our Editorial version 1.2:

https://www.geosci-model-dev.net/12/2215/2019/

This highlights some requirements of papers published in GMD, which is also available

on the GMD website in the 'Manuscript Types' section:

http://www.geoscientific-model-development.net/submission/manuscript_types.html

In particular, please note that for your paper, the following requirement has not been met in the Discussions paper:

- "The main paper must give the model name and version number (or other unique identifier) in the title."

Please add the version number for ROMS(v3.6) to the title upon your revised submission to GMD.

Yours,

Astrid Kerkweg
* * *

---

## Author Comment (AC1) · 28 Mar 2020

Thank you for the clarification, we will add the ROMS version number for the revised submission.

With thanks, Alexander Lopez

---

## Referee Comment (RC1) · Anonymous Referee #1 · 8 Apr 2020

General Comments

This manuscript describes the development and evaluation of a regional modeling system (Doppio) for simulating the circulation and hydrography of the shelf/slope region off the northeast United States. As the authors note, regional oceanographic models like this are useful in that their results can provide the physical underpinning for studies of coastal ocean biogeochemistry and ecosystem function, and can themselves be used to investigate interannual and long-term variability in shelf circulation and hydrography.

[Figure]

The Doppio modeling system is based on the Regional Ocean Modeling System (ROMS), which is well described in the oceanographic literature, thus allowing the authors to dispense with detailed descriptions of model parameterizations and numerical schemes. Their focus is thus on the description of the external forcing (tides, river inflows, meteorological forcing, and open boundary forcing) and on the evaluation of the model results in comparison with numerous observational datasets (water properties, currents, sea surface elevation). The authors do a good job in describing the development and testing of the external forcing and the tradeoffs that are needed. The comparisons of model output and observational data are wide-ranging and give a clear picture of the model fidelity.

Overall, I consider this manuscript to be well written and worthy of publication. However, I have a number of specific comments, noted below, that I believe should be addressed.

I found the description of the river inflow forcing to be somewhat unclear. As best I can tell, the authors use a statistical approach whereby the integrated discharge over fairly large regions are predicted using gauged discharge at 27 of the largest rivers. The integrated discharge dataset comes from the Stewart et al (2013) reference. Examination of the list of the 27 rivers indicates that almost all are large rivers, with the exception of the Quashnet (which I note is NOT located in New Hampshire as indicated in the figure 1 legend) and Carmans. From the USGS website, the Quashnet (located on Cape Cod) has a tiny mean discharge of about 20 ftˆ3/s and a drainage area of only 2.6 square miles. It seems questionable to use such a small stream to predict the discharge of rivers over a wide area (it is not clear from the manuscript how large an area is represented by this river source). I would suggest the addition of a bit more detail in this section of the manuscript in order to flesh out some of these details. I also note that I was unable to locate the USGS page for the Carmans River, so it is not clear to me where the authors are obtaining its discharge. I know that this is a very small stream as well. My guess is that it is used to predict the discharge of all rivers on Long

Island, none of which are particularly large, so this may not be a bad choice.

Specific Comments lines 79-81: The characterization of the GOM's bathymetry and currents as "uncommon" seems a bit strange. What exactly is uncommon? Granted that the bathymetry is rougher than that of the MAB shelf and the tidal currents are stronger, but uncommon is not a useful descriptor in my view.

last paragraph of section 1: This may seem like a minor issue, but is it necessary to use words like "hibernal" and "estival" instead of wintertime and summertime? While it is nice to learn new words, having to look them up does interrupt the flow of reading.

Figure 1: the legend for the moorings indicates (bold type face) that moorings B, I, and N are used for later correlation analysis, but the text (sect. 3.4) says that moorings B, N, and M are used.

Figure 2b: It looks like the estimates of PAR from corrected NARR shortwave radiation are biased low relative to MODIS PAR (many points below the 1-1 line). At least by eye, it looks like a regression line would lie well below the 1-1 line. So, I don't understand why the authors state that the mean ratio is close to unity.

Lines 215-220: The description of the open boundary bias correction is not clear to me. I think that they adjust the mean of the boundary variables to match the mean values from their local analysis, however the writing is vague.

Figure 7c and 7d: The symbols in these figures are so numerous and overlapping that it is hard to decipher them. It is difficult to see the increase in correlation using the bias-corrected boundary conditions that the authors describe in lines 295-300 of the text.

Figure 8: It looks like the coherence (blue) is only plotted for cases where the lower bound of the 90% confidence interval is greater than zero. This should be stated in the caption. Alternatively, the figure could be revised to include only the coherence and the level of significant coherence. This would make the figure less busy and allow more

detail to be seen.

line 210: Is it correct to say that at site N, the model captures high frequency and seasonal timescales? The coherences are 0.3 and lower, so the correspondence is not very high.

Figure 15: I think that some discussion of the differences between Doppio and the 4Dvar climatology over the outer shelf on Georges Bank and the MAB is needed. Doppio shows a separation between the coastal flow and the shelfbreak flow whereas the 4Dvar field indicates equatorward flow over the outer shelf as well. Why is this so?

Figure 16: The bottom figure is supposed to represent the flow in the region from 100m depth to the sea floor. Why are there arrows plotted on the MAB shelf where water depth is less than 100 m?

[Figure]

---

## Referee Comment (RC2) · Anonymous Referee #2 · 27 May 2020

General Comments

The authors clearly present the details of a new regional ocean model configuration, for the coastal, shelf and slope waters of the Mid-Atlantic Bight and Gulf of Maine – of broad interest to biological and biogeochemical applications in particular. With this in mind, the authors could refer a little more to the importance of the physics and dynamics – painstakingly evaluated – for these applications. For example, water depth shoals to just a few metres across part of Georges Bank, and a large expanse of the outer shelf here (around 200 x 100 km) is no deeper than 60 m. These shallow waters

remain tidally mixed all year around, while the deeper surrounding waters thermally stratify during summer. The consequence is enclosure of mixed waters by a tidal mixing front, which supports a clockwise residual gyre (Chen et al. 2003). I return to this in Specific Comments below.

Reference: Chen, C., Beardsley, R.C., Franks, P.J.S., and J. Van Keuren (2003). Influence of diurnal heating on stratification and residual circulation of Georges Bank, J. Geophys. Res., 108, 8008, doi:10.1029/2001JC001245, C11.

The manuscript should nevertheless be suitable for publication in GMD, subject to minor and technical revisions in response to the following comments.

Specific Comments

1. p.3, lines 100-103: Relate the horizontal resolution (& km) to the 1st barocinic Rossby radius in shallow stratified water at mid-latitudes, and discuss how much of the shallow dynamics is unresolved

2. p.9, lines 320-321: The water column must be stabilized in winter by salinity stratification, as noted below (p.10, lines 338-339) – this should be introduced here to avoid an impression of static instability.

3. p.10, line 352: As outlined above, water columns on Georges Bank remain fully mixed during summer. Surrounded by stratified water, horizontal density gradients across the front between mixed and stratified water support a baroclinic jet (Chen et al. 2003), presumably of consequence for pelagic and benthic fauna. How well is this simulated in Doppio?

4. Fig. 10: the indication is of seasonal stratification across Georges Bank, presumably due to area-averaging? In general, averaging across the large areas in Fig. 6 (left) will "hide" considerable spatial structure in seasonal stratification – can you justify this averaging?

5. Figs. 15, 16: Seasonal jets will not be clearly seen in long-term averages – can you

contrast winter and summer circulation, zooming in on Georges Bank? (an additional figure)

Technical Corrections

1. p.7, line 223: "reference datum"

2. p.13, line 441: "in the face of"

3. Figure 6: explain the dashed contours (labelled 0.5, 1.0)

---

## Author Comment (AC2) · 12 Jun 2020

(Reviewer 1 without emphasis, Author response to comments with bold emphasis)

General Comments
This manuscript describes the development and evaluation of a regional modeling system (Doppio) for simulating the circulation and hydrography of the shelf/slope region off the northeast United States. As the authors note, regional oceanographic models like this are useful in that their results can provide the physical underpinning for studies of coastal ocean biogeochemistry and ecosystem function, and can themselves be used to investigate interannual and long-term variability in shelf circulation and hydrography.

The Doppio modeling system is based on the Regional Ocean Modeling System (ROMS), which is well described in the oceanographic literature, thus allowing the authors to dispense with detailed descriptions of model parameterizations and numerical schemes. Their focus is thus on the description of the external forcing (tides, river inflows, meteorological forcing, and open boundary forcing) and on the evaluation of the model results in comparison with numerous observational datasets (water properties, currents, sea surface elevation). The authors do a good job in describing the development and testing of the external forcing and the tradeoffs that are needed. The comparisons of model output and observational data are wide-ranging and give a clear picture of the model fidelity.
Overall, I consider this manuscript to be well written and worthy of publication. However, I have a number of specific comments, noted below, that I believe should be addressed.

I found the description of the river inflow forcing to be somewhat unclear. As best I can tell, the authors use a statistical approach whereby the integrated discharge over fairly large regions are predicted using gauged discharge at 27 of the largest rivers. The integrated discharge dataset comes from the Stewart et al (2013) reference. Examination of the list of the 27 rivers indicates that almost all are large rivers, with the exception of the Quashnet (which I note is NOT located in New Hampshire as indicated in the figure 1 legend) and Carmans. From the USGS website, the Quashnet (located on Cape Cod) has a tiny mean discharge of about 20 ft^3/s and a drainage area of only 2.6 square miles. It seems questionable to use such a small stream to predict the discharge of rivers over a wide area (it is not clear from the manuscript how large an area is represented by this river source). I would suggest the addition of a bit more detail in this section of the manuscript in order to flesh out some of these details. I also note that I was unable to locate the USGS page for the Carmans River, so it is not clear to me where the authors are obtaining its discharge. I know that this is a very small stream as well. My guess is that it is used to predict the discharge of all rivers on Long Island, none of which are particularly large, so this may not be a bad choice.
- **Quashnet location corrected in Fig. 1 legend**
- **USGS site 01306460 is used for Carmans. At the time of model development, the gauge was labeled "CONNETQUOT (Carmans R) BK NR CENTRAL ISLIP NY."**
- **The USGS discharge for the Carmans is scaled by the watershed discharge for all points on Long Island, NY, as such it serves as a collective for all of the relatively small rivers on Long Island and sources them to the model at one location.**

Specific Comments lines 79-81: The characterization of the GOM's bathymetry and currents as "uncommon" seems a bit strange. What exactly is uncommon? Granted that the bathymetry is rougher than that of the MAB shelf and the tidal currents are stronger, but uncommon is not a useful descriptor in my view.

- **Word choice has been changed. Author's intent was only to note GOM bathymetry was directly shaped by glaciers and thus a more varied profile compared to the MAB.**

last paragraph of section 1: This may seem like a minor issue, but is it necessary to use words like "hibernal" and "estival" instead of wintertime and summertime? While it is nice to learn new words, having to look them up does interrupt the flow of reading.
- **Word choice has been changed.**

Figure 1: the legend for the moorings indicates (bold type face) that moorings B, I, and N are used for later correlation analysis, but the text (sect. 3.4) says that moorings B, N, and M are used.
- **Figure 1 legend has been corrected.**

Figure 2b: It looks like the estimates of PAR from corrected NARR shortwave radiation are biased low relative to MODIS PAR (many points below the 1-1 line). At least by eye, it looks like a regression line would lie well below the 1-1 line. So, I don't understand why the authors state that the mean ratio is close to unity.
- **Author acknowledges the corrected NARR shortwave radiation to be biased low relative to MODIS PAR when PAR is above 45, however this is only for two-thirds of the observations. The remaining third is the opposite, resulting in the mean ratio to be near 1.**

Lines 215-220: The description of the open boundary bias correction is not clear to me. I think that they adjust the mean of the boundary variables to match the mean values from their local analysis, however the writing is vague.
- **This interpretation is correct; we will rephrase the writing to better clarify the method.**

Figure 7c and 7d: The symbols in these figures are so numerous and overlapping that it is hard to decipher them. It is difficult to see the increase in correlation using the bias-corrected boundary conditions that the authors describe in lines 295-300 of the text.
- **Figure has been reworked so it is more legible.**

Figure 8: It looks like the coherence (blue) is only plotted for cases where the lower bound of the 90% confidence interval is greater than zero. This should be stated in the caption. Alternatively, the figure could be revised to include only the coherence and the level of significant coherence. This would make the figure less busy and allow more detail to be seen.
- **Further explanation will be added to the caption.**

line 310: Is it correct to say that at site N, the model captures high frequency and seasonal timescales? The coherences are 0.3 and lower, so the correspondence is not very high.
- **While the coherences 0.3 are low, they are still statistically significantly not zero. We will add text clarifying and acknowledging this.**

Figure 15: I think that some discussion of the differences between Doppio and the 4Dvar climatology over the outer shelf on Georges Bank and the MAB is needed. Doppio shows a separation between the coastal flow and the shelfbreak flow whereas the 4Dvar field indicates

equatorward flow over the outer shelf as well. Why is this so?

- **The 4D-Var climatology solution is computed from a model without tides and some other high frequency dynamics that may lead to rectified flows. Without further analysis we cannot easily explain the cause of these differences. We have noted in the revised text the reviewer's observation of these differences for completeness. Of note is that the latest CNES-CLES18 MDT (included below) has a similar feature to that which reviewer 1 describes, which indicates further that the true pattern is uncertain.**

[Figure]

Figure 16: The bottom figure is supposed to represent the flow in the region from 100m depth to the sea floor. Why are there arrows plotted on the MAB shelf where water depth is less than 100 m?

- **This description has been corrected to model surface and bottom layers, hence the arrows plotted in the less than 100 m waters.**

---

## Author Comment (AC3) · 12 Jun 2020

(Reviewer 2 without emphasis, Author response to comments with bold emphasis)

**General Comments**

The authors clearly present the details of a new regional ocean model configuration, for the coastal, shelf and slope waters of the Mid-Atlantic Bight and Gulf of Maine – of broad interest to biological and biogeochemical applications in particular. With this in mind, the authors could refer a little more to the importance of the physics and dynamics – painstakingly evaluated – for these applications. For example, water depth shoals to just a few metres across part of Georges Bank, and a large expanse of the outer shelf here (around 200 x 100 km) is no deeper than 60 m. These shallow waters remain tidally mixed all year around, while the deeper surrounding waters thermally stratify during summer. The consequence is enclosure of mixed waters by a tidal mixing front, which supports a clockwise residual gyre (Chen et al. 2003). I return to this in Specific Comments below.

**- Authors have noted this in the brief overview of regional dynamics in Section 1.**

Reference: Chen, C., Beardsley, R.C., Franks, P.J.S., and J. Van Keuren (2003). Influence of diurnal heating on stratification and residual circulation of Georges Bank, J. Geophys. Res., 108, 8008, doi:10.1029/2001JC001245, C11.

The manuscript should nevertheless be suitable for publication in GMD, subject to minor and technical revisions in response to the following comments.

**Specific Comments**

1. p.3, lines 100-103: Relate the horizontal resolution (& km) to the 1st barocinic Rossby radius in shallow stratified water at mid-latitudes, and discuss how much of the shallow dynamics is unresolved

**- Authors now note relation to 1st baroclinic Rossby radius for shelf waters in text.**

2. p.9, lines 320-321: The water column must be stabilized in winter by salinity stratification, as noted below (p.10, lines 338-339) – this should be introduced here to avoid an impression of static instability.

**- We have noted the role of salinity in maintaining stability.**

3. p.10, line 352: As outlined above, water columns on Georges Bank remain fully mixed during summer. Surrounded by stratified water, horizontal density gradients across the front between mixed and stratified water support a baroclinic jet (Chen et al. 2003), presumably of consequence for pelagic and benthic fauna. How well is this simulated in Doppio?

- Shown below are 5 transections for the mean of July '08, which depict a mixed water column within the shallower portion Georges Bank. The dotted line(s) on the vertical transects represent the intersection with the other transect(s). Model shows vertically well-mixed water column at shallower locations on Georges Bank for the sample month of July '08. Surface velocity is shown below in response to Specific Comment #5.

4. Fig. 10: the indication is of seasonal stratification across Georges Bank, presumably due to area-averaging? In general, averaging across the large areas in Fig. 6 (left) will "hide" considerable spatial structure in seasonal stratification – can you justify this averaging?

- The model subregions described in Fig. 6 were chosen to have enough observations in each region to have meaningful model-data comparison statistics, while also being relatively consistent dynamical regimes.

5. Figs. 15, 16: Seasonal jets will not be clearly seen in long-term averages – can you contrast winter and summer circulation, zooming in on Georges Bank? (an additional figure)

- Seasonal jets are visible in the plot of monthly mean surface current velocity (m s-1), shown below, for July 2008 and January 2009. Bathymetry contours of 100, 200, and 4000 m are shown in white. The model domain encompasses many different dynamical regions of interest to researchers, but we do not feel we have the space in this overview paper to add numerous figures depicting regions of particular interest. We have, however, made all model output openly accessible to the community from whom we invite further analysis.